# Graduated Compression Stockings for Thromboprophylaxis in Orthopaedic and Trauma Surgery: A Rapid Review and Meta-Analysis

**DOI:** 10.3390/jcm14238578

**Published:** 2025-12-03

**Authors:** Dirk Stengel, Daniela Schnorbus, Axel Ekkernkamp, Matthias Münzberg, Beate Schmucker, Lina El Kassar, Flemming Rohrmann, Paul A. Grützner

**Affiliations:** 1BG Kliniken—Klinikverbund der Gesetzlichen Unfallversicherung gGmbH, Leipziger Pl. 1, 10117 Berlin, Germany; daniela.schnorbus@bg-kliniken.de (D.S.); axel.ekkernkamp@ukb.de (A.E.); beate.schmucker@bg-kliniken.de (B.S.); lina.elkassar@bg-kliniken.de (L.E.K.); flemming.rohrmann@bg-kliniken.de (F.R.); paul.gruetzner@bgu-ludwigshafen.de (P.A.G.); 2BG Klinikum Unfallkrankenhaus Berlin gGmbH, Warener Str. 7, 12683 Berlin, Germany; 3BG Unfallklinik Frankfurt am Main, Friedberger Landstr. 430, 60389 Frankfurt am Main, Germany; matthias.muenzberg@bgu-frankfurt.de; 4BG Klinik Ludwigshafen, Ludwig-Guttmann-Str. 13, 67071 Ludwigshafen, Germany

**Keywords:** thromboembolism, prophylaxis, orthopedics, arthroplasty, spine, pelvis, trauma, fracture, compression stockings, heparins, fondaparinux

## Abstract

**Background/Objectives:** The utility and value of graduated compression stockings (gCS) as an adjunct to pharmacological thromboprophylaxis, with and without low-molecular-weight heparins (LMWH) and other anticoagulants, in avoiding any thromboembolic (TE) event in the scenario of total joint replacement, fracture management, spine and pelvic surgery, and arthroscopic procedures, remains unclear. Because of the urgent need to decide whether gCS should stay in the portfolio of a national group of nine tertiary trauma centres, our research department was requested to answer the question of whether gCS provide any extra benefit in addition to modern TE prophylaxis in orthopaedic and trauma surgery through a prospectively registered rapid review (PROSPERO CRD42024621104). **Methods:** We searched PubMed, Ovid MEDLINE, Embase, CINAHL, and CENTRAL from 1 January 1980 to 1 March 2025, for randomised controlled trials (RCTs) and cohort studies comparing TE prophylaxis regimens, both with and without gCS, and modern pharmacological anticoagulants. The methodological quality of individual studies was rated by the Cochrane Collaborations’ Risk of Bias Version 2.0 (RoB-2) and the Risk of Bias in Non-Randomised Studies of Interventions (ROBINS-I) tools, supplemented by the Grading of Recommendations Assessment, Development and Evaluation (GRADE). The reported cumulative incidence of any TE event (i.e., deep vein thrombosis, pulmonary embolism), as defined by individual trialists, was chosen as the primary endpoint, and expressed as the relative risk (RR) between intervention and control groups. **Results:** Fifteen investigations (13 RCTs and 2 observational studies) enrolling 7721 patients (mean age, 59 [SD 13] years; 3538 males [46%]) with various musculoskeletal conditions and injuries were included. Methodological quality was deemed sufficient to derive meaningful conclusions. The random-effects pooled RR across all studies was 1.15 (95% confidence interval [CI]: 0.80–1.64) in favour of the no-gCS control, but with substantial heterogeneity (I^2^: 73%). Only three studies investigated the effectiveness of gCS versus no prophylaxis (*N* = 246, RR: 0.72, 95% CI: 0.43–1.22). Seven studies (*N* = 5117) compared various combinations of pharmacological prophylaxis, with or without gCS, for a summary RR of 1.44 (95% CI: 0.76–2.72). **Conclusions:** The results of this rapid review neither show a clear benefit nor support the general use of gCS to prevent TE in orthopaedic and trauma surgery, especially if pharmacological prophylactic measures are established and suitable.

## 1. Introduction

Patients undergoing elective orthopaedic procedures (e.g., total joint arthroplasty, arthroscopy, and spinal interventions) or admitted to the hospital due to injuries and fractures are at an increased risk of thromboembolism (TE), including deep vein thrombosis (DVT) and pulmonary emboli (PE), because of immobilisation and impaired function of the calf pump [1]. The additional risk caused by surgery itself (activation of coagulation cascades, second-hit inflammatory response, etc.) is challenging to quantify. Obviously, there must be a difference in the risk of TE between elective and trauma patients, as well as differences due to injury severity, demographic factors, comorbidity, smoking, medication, and so on.

The reported incidence of TE events also depends on the diligence of diagnosis and the effectiveness of screening protocols. Partial verification bias may occur if symptomatic TE is confirmed only through sonographic or computed tomography imaging. Realistically, TE may affect 1–10% of patients in this setting [2,3,4,5,6,7,8,9].

Pharmacological prophylaxis with low-molecular-weight heparins (LMWH) or factor Xa inhibitors, such as fondaparinux, remains the dominant strategy in this scenario [10].

Aspirin (acetylsalicylic acid) has recently emerged as the preferred drug for TE prophylaxis in the United States and other countries, likely because it is inexpensive and available over the counter without a prescription. Yet, as an irreversible inhibitor of cyclooxygenases (COX) 1 and 2, the clinical effectiveness of aspirin in this setting (as shown by the PREVENT-CLOT trial [11]) is difficult to explain through its pharmacodynamic action and targets, which do not interfere with intrinsic coagulation cascades [12].

Graduated compression stockings (gCS) have long been used as a basic physical prophylactic measure in surgery (when no other TE prophylaxis is available), under the assumption that they passively support the calf pump. However, they may be inconvenient for both patients and providers, as they tend to slip and slide, thereby negating any intended effect. There is ongoing doubt about the added value of gCS to modern pharmacological thromboprophylaxis protocols. Still, the external validity and transferability of previous meta-analyses to orthopaedic and trauma surgery are difficult to estimate.

A meta-analysis by Milinis et al. already found a lack of head-to-head comparisons between extended pharmacological DVT prophylaxis and gCS alone, with the three trials that included only gCS focusing on abdominal surgery [13]. In a network meta-analysis, Wong et al. included only a single trial using gCS in a review of 14 studies including Asian patients undergoing total knee arthroplasty [14]. Finally, Dave et al. conducted a systematic review of prophylactic measures after knee arthroscopy, without data aggregation [15].

In times of an information explosion, ever-decreasing healthcare resources, and limited time to make decisions, “classic” systematic reviews (specifically Cochrane Reviews) on old trials cannot provide quick answers to an acute strategic problem, which may entail substantial direct costs. Rapid reviews are sometimes necessary in such situations, providing a “go” or “no go” decision in favour of or against a particular intervention [16]. A Cochrane Rapid Reviews Methods Group guidance paper was published on this topic [17].

At the request of the strategic purchasing and medical control departments of our hospital group, we set out to complement and provide more detail on previous evidence through a rapid review and meta-analysis of randomised controlled trials and cohort studies, aiming to answer the following PICOT (Patient/Population, Intervention, Comparison, Outcome, Timeframe) question:

In patients of any age and gender with any injury or fracture, or those undergoing any elective orthopaedic or spine procedure, does the prescription of gCS lower the risk of TE as compared to any control (e.g., no prophylaxis, sham or placebo, LMWH, fondaparinux, aspirin) at any time of follow-up?

## 2. Materials and Methods

This rapid review and meta-analysis, prospectively registered in PROSPERO (CRD42024621104), aimed to evaluate the effectiveness of gCS compared to any control regimen in preventing TE in patients undergoing elective orthopaedic procedures or those requiring surgical care for musculoskeletal or general trauma.

### 2.1. Search Methods

We searched for RCTs and cohort studies (with and without propensity score-matching) comparing TE prophylaxis with and without gCS in the setting of orthopaedic and trauma surgery. This included elective total joint replacement, fracture care, spine procedures, arthroscopic surgery, and management of multiple injuries. No restrictions were established for publication date or language. The latest search was conducted on 1 March 2025. We searched PubMed, Ovid Medline, Embase, CINAHL (via EBSCOhost), and the Cochrane Library and CENTRAL databases for previous systematic reviews and individual studies. This was followed by a snowball procedure that employed related articles and references cited in individual manuscripts, as well as a generic Google search. We placed no restrictions on time or language.

### 2.2. Primary Endpoint

The primary endpoint was the cumulative number or rate of any thromboembolic event at the latest time of follow-up, as reported by the authors. This was considered a patient-relevant, critical outcome for clinical decision-making in the Grading of Recommendations Assessment, Development and Evaluation (GRADE) hierarchy.

### 2.3. Secondary Endpoints

Secondary endpoints included proximal deep vein thrombosis (DVT) and/or pulmonary embolism (PE), overall mortality, and adverse events (AE), including serious adverse events (SAE), with the use of gCS. If reported, we considered patient-centred outcomes, such as health-related quality of life or function.

### 2.4. Data Abstraction

Data from full texts and electronic appendices were abstracted by one reviewer (Di.S.) and verified by two others (Da.S., F.R.) for numerical accuracy. We primarily extracted reported cumulative rates of any TE, as defined by the authors, irrespective of the diagnostic modality (e.g., Doppler–Duplex ultrasonography, venography), whether used in a screening or in a confirmatory approach. Screenings involved routinely examining all patients after surgery for the presence of venous clots, regardless of physical signs, thereby minimising the risk of false-negative results. Confirmatory studies were performed to verify symptoms (swelling, pain, shortness of breath) suggestive of TE. We also recorded baseline demographics (i.e., age and gender), the clinical condition or scenario of interest, and assessed and grouped control interventions (e.g., no prophylaxis, mechanical, pharmacological).

### 2.5. Grading of Methodological Quality

The Cochrane Risk of Bias Version 2.0 (RoB-2) instrument and traffic light system were used to assess the methodological quality of the RCTs [18]. The rating was primarily abstracted the data. Doubtful situations were rare and typically resolved through discussion among team members with specific clinical experience. We assigned a “some concerns” rating to studies that used sealed envelopes for randomisation (as they can be easily breached, unlike telephone- or web-based randomisation tools) and those that lacked a trial flow diagram. While checklists like the Consolidated Standards of Reporting Trials (CONSORT) [19] have only become mandatory in recent years, they enable the identification of bias due to deviations from the original treatment assignment (i.e., unintended crossovers), missing data, and selective reporting. In the absence of such a unified scheme, we rated the RoB-2 criteria as having “some concerns” to maintain fairness.

Similarly, the Cochrane Risk Of Bias In Non-Randomised Studies of Interventions (ROBINS-I) tool was used to assess the methodological quality of observational studies [20]. A “serious risk of bias” was assumed if imaging for TE was only performed in the case of symptoms. We used the RobVis online visualisation tool, developed by the Bristol group, to create individual traffic light and summary plots (https://www.riskofbias.info/welcome/robvis-visualisation-tool, accessed on 15 March 2025) [21].

We also provided a summary table based on the GRADE scheme, a systematic framework for assessing the strength of evidence supporting healthcare recommendations [22]. This quantifies whether the estimated effect of an intervention accurately reflects its actual impact in the real world (https://methods.cochrane.org/gradeing/, accessed on 1 October 2025). With hundreds of publications and handbooks on the development of GRADE and its practical application, we found this research note [23] and webpage illustrative: https://www.uniqcret.com/post/grade-certainty-evidence, accessed on 1 October 2025. Based on five domains (i.e., risk of bias, inconsistency, indirectness, imprecision, and publication bias), we rated the certainty of the current evidence as high (very confident that the actual effect is close to the estimate), moderate (probably near), limited (actual effect may be substantially different), or very low (very little confidence in the forecast). We used the Cochrane GRADE pro GDT platform (https://methods.cochrane.org/gradeing/gradepro-gdt, accessed on 1 October 2025) as an orientation tool.

### 2.6. Statistical Analysis

For primary endpoint analysis, we did not distinguish between superficial and deep vein thromboses or PE; we regarded a venous clot as a venous clot, with all its potential local and systemic clinical consequences. We also accepted any definition of TE by individual trialists and summarised all events as a composite outcome. Given the lack of granularity and the risk of misclassification bias, we considered this approach clinically reasonable. However, we conducted separate and subgroup analyses of studies that used screening Doppler–Duplex ultrasonography or venography, regardless of the presence of clinical signs and symptoms. Further subgroup and sensitivity analyses comprised the following: 1. patients undergoing total joint arthroplasty, 2. RCTs and propensity score-matched cohort studies (we considered propensity score-matching a quasi-experimental approach, allowing for causal inferences close to RCTs), and 3. studies conducted only after 1990 and/or those not employing outdated pharmaceuticals like Dextran. Proximal DVT and/or PE, as well as all-cause mortality at any time, were analysed secondarily, as this was specified in the individual studies. Additionally, we sought to report adverse events (AEs) and serious adverse events associated with gCS use.

To ease readability and make this work precise and focused, we adhered to the following statistical analysis and reporting plan:The trial profile was tabulated in detail, including sample sizes, demographic information (i.e., age and gender), the condition of interest (i.e., general trauma, fractures, total joint arthroplasty, arthroscopy, spine surgery, and others), individual interventions, and primary and secondary endpoints.Unweighted individual risk ratios (RR) and risk differences (RD) were illustrated by forest plots for each study and each comparison (as some trials had three or four arms or comparators, e.g., gCS only, LMWH, Fondaparinux, and placebo, and even intermittent pneumatic compression or a combination of methods), and for both the primary endpoint (i.e., any TE), and the secondary endpoint of proximal DVT and/or PE. This was carried out to provide readers with a comprehensive overview of the distribution of effects.A complete random-effects meta-analysis using RR as the effect size was conducted only on the full sample of trials and studies for the primary endpoint of any TE. This included forest plots of individual and pooled RR with 95% confidence intervals (CI), funnel plots to illustrate potential publication bias, and Egger’s tests. We reasoned that for both clinically and healthcare system-relevant decisions, this was the most crucial analysis. We added pooled RR estimates for secondary endpoints and RD estimates to calculate the Numbers Needed to Treat (NNT) without further analyses of possible publication bias.The results from the subgroup mentioned above and the sensitivity analyses were reported in a tabular format.

We reasoned that this targeted, priority-oriented approach, complying with the idea of a rapid review, would be more concise and valuable, especially to clinical readers and other busy stakeholders, than an abundance of tables and graphics. We would be more than happy to either conduct additional analyses ourselves to answer specific questions or to share our anonymised dataset with other researchers upon request.

The meta package is embedded in STATA 18.0 (StataCorp, Armonk, NY, USA). I^2^ was employed to determine heterogeneity across studies. We considered network meta-analysis, but this did not allow for meaningful or sufficiently large group formation in this specific scenario.

### 2.7. Use of AI

We used the Grammarly language writing assistant tool (Superhuman Platform Inc., San Francisco, CA, USA), to enhance style and grammar. This manuscript was written by human experts only. Additionally, all data collection from published manuscripts, as well as data abstraction, analysis, and presentation, were performed by human beings.

## 3. Results

### 3.1. Search Results, Risk of Bias, and Study Profile

The search strategy and study selection process are illustrated in the PRISMA 2020 flow diagram (Figure 1), extended by the PRISMA checklist and retrieval results from individual databases in Appendix A (Table A1, Table A2, Table A3 and Table A4) and Appendix B. Altogether, we identified 15 investigations (13 RCTs [24,25,26,27,28,29,30,31,32,33,34,35,36] and 2 cohort studies [37,38]) meeting our eligibility criteria. The studies were published between 1978 and 2021. They included 7721 eligible patients with various musculoskeletal conditions and injuries (with slightly smaller numbers depending on individual analyses, losses to follow-up, missing data, etc.).

The largest RCT to date addressing our PICOT question is the GAPS (Graduated compression as an Adjunct to thrombo-Prophylaxis in Surgery) non-inferiority RCT commissioned by the UK National Institute for Health and Care Excellence (NICE) [33,39]. Yet, with 940 patients in the gCS plus LMWH group and 948 in the LMWH group, it enrolled only 18 and 39 participants, in the orthopaedic and plastic procedures group. If another 62 neurosurgical interventions had matched our target population, the trial would have enrolled only 6.8% of patients who could answer our primary objective. The large-scale (*N* = 798), propensity score-matched cohort study by Arabi et al. [34] also included only 278 (34.8%) patients relevant to our clinical scenario.

Nine trials investigated the utility of gCS in total joint arthroplasty [24,26,27,28,30,31,32,35,36]. Two of them were published before 1990 [24,32] and used either no other prophylactic measures or outdated pharmaceuticals (e.g., intravenous Dextran-70).

Partial verification bias occurs when the diagnostic tests used to confirm TE are performed only in the presence of clinical signs or symptoms, or when only positive results of imperfect tests (e.g., clinical decision rules) are confirmed by imaging. Eleven studies used a screening protocol [24,25,26,27,28,29,30,31,33,34,35], and we performed a subgroup analysis using only these studies. Reporting of individual TE events (i.e., DVT, pulmonary embolism) varied across studies, hampering valid subgroup analyses.

The same was true for AE, SAE, and mortality. We therefore decided to focus on the primary outcome addressed and reported by all studies—the cumulative incidence of any TE event with and without gCS.

Table 1 sketches the certainty of evidence according to GRADE. Table 2 provides a comprehensive summary of the key criteria of the included RCTs and cohort studies, the demographic baseline profiles of participants, the intervention and control regimens, and the methods used to detect TE.

It became clear to us that “head-to-head” comparisons of gCS with no prophylaxis or with obsolete approaches, such as intravenous dextran-70, were almost exclusively based on historical trials. In more recent studies, gCS were reported or considered (if at all) only as a co-therapy. We used any information on a possible additional benefit of gCS that we could extract from scientific publications (including electronic supplementary material). This may have required a reversal of intervention and control groups.

We were confronted with the following questions:Are gCS superior to no prophylaxis in preventing TE (a situation that is almost absent today)?Are gCS, as a mechanical method, not inferior to pharmacological TE prophylaxis (we considered this to be unlikely based on the data and clinical experience)?Do gCS provide any additional benefit to anticoagulants in reducing the risk of TE?

Altogether, the methodological quality of both experimental and observational research was deemed adequate to derive meaningful conclusions from the available evidence. Table 1 illustrates the certainty of the evidence according to GRADE. Table 2 sketches individual trial, patient, condition, and intervention profiles. Figure 2 and Figure 3 display the results of RoB-2- and ROBINS-I assessments.

### 3.2. Quantitative Findings

Figure 4 and Figure 5 illustrate the distributions of unweighted RR and RD across the interventional and control groups for the primary endpoint of any TE event. Figure 6 and Figure 7 present distributions for proximal DVT and PE.

A random-effects meta-analysis yielded a pooled overall RR of 1.15 (95% CI: 0.80–1.64) (Figure 8), showing no evidence for or against the overall effectiveness of gCS. There was no evidence of publication bias or small study effects (Figure 9).

The associated RD was 0.90% (95% CI: −0.45–2.24%). As this is compatible with chance, no NNT should be calculated based on these numbers.

We identified three main clusters of comparisons: (1) gCS versus any LMWH or Fondaparinux, (2) gCS versus any Other or Combined Regimens, and (3) gCS versus no prophylaxis. The second formed a heterogeneous subgroup, but this was considered the best approach to bundle all data in a reasonable way (Figure 10).

The study by Yokote et al. [35] shared features of clusters (1) and (2). When moving this study to cluster (2) in a sensitivity attempt, the results did not markedly change. Pooled RR changed from 1.44 (95% CI: 0.76–2.72) to 1.52 (95% CI: 0.72–3.21), and from 1.17 (95% CI: 0.73–1.86) to 11.6 (95% CI: 0.80–1.69).

Ten studies with 6496 participants reported proximal DVT and PE, with a pooled RR of 1.17 (95% CI: 0.62–2.21) against gCS. The related pooled RD was 0.2% (95% CI: −0.4–0.8%).

Detailed results from subgroup and sensitivity analyses are shown in Table 3.

### 3.3. Further Analyses

Mortality was infrequently reported. No deaths were observed in the trial by Camporese et al. [25]. The upper 95% confidence limits for null events in the gCS group (0/660) and the combined LMWH groups (0/1101) are 0.6% and 0.3%, respectively. Fuji et al. [28] also did not observe deaths, resulting in an upper 95% confidence limit of 1.8%.

In the GAPS trial, the rates of death in the LMWH, LMWH plus gCS, and gCS groups were 2/779, 0/787, and 0/160, respectively [33]. It was impossible to break this down to the orthopaedic and/or trauma subgroups.

Finally, Arabi et al. reported propensity-score-adjusted hazard ratios (HRs) for hospital mortality with gCS compared to no prophylaxis (0.86, 95% CI: 0.62–1.21).

No reliable information on health-related quality of life or functional scores could be found in individual investigations, which adds to our PICOT question and specific scenario. Also, while pharmacological trials of Enoxaparin, Edoxaban, Dalteparin, and/or Fondaparinux reported bleeding complications, etc., no information was provided on AE or SAE with gCS.

## 4. Discussion

### 4.1. General Findings

This rapid review and meta-analysis found no convincing evidence that graduated compression stockings (gCS) add a measurable benefit to current pharmacological prophylaxis protocols using LMWH, Fondaparinux, etc., for reducing the risk of thromboembolic (TE) events in elective orthopaedic procedures, orthopaedic trauma, or system injuries.

Respecting the logical principle “absence of evidence does not mean evidence of absence”, we must consider “evidence of absence” here using the best available evidence. The estimates in any subgroup or sensitivity analysis were compatible with chance.

There may still be indications for applying gCS to individual patients (e.g., those who are difficult or impossible to mobilise after surgery, or subjects with any contraindication for pharmacological prophylaxis). In this case, however, intermittent pneumatic compression (IPC) as an active measure to stimulate the calf pump may be a more effective mechanical TE prophylaxis [40].

There was no evidence of publication bias, and no novel, large-scale confirmatory trial can ever be conducted comparing gCS only with no or pharmacological TE prophylaxis in the scenario of interest, as that would contradict the recommendations of current evidence-based clinical practice guidelines [41,42].

### 4.2. Limitations

The most significant drawback of this work is its heterogeneity—there are few studies with precisely defined, unequivocally planned intervention and control groups that incorporate gCS. In addition, differences in pharmacological protocols, methods for detecting thromboembolism, study eras, and various surgical conditions contributed to the dispersion of effect sizes.

We made a considerable effort to analyse the additional effectiveness of gCS across a broad range of interventions but encountered limitations associated with reporting by individual publications. The various combinations of pharmacological prophylactic regimens across different settings and populations led to a nearly uncountable number of possible interactions.

Partial verification bias may play a substantial role when considering TE. Whether institutions implement a screening strategy and conduct this during clinical trials (e.g., through regular Doppler–Duplex ultrasonography of the deep veins), or react to the physical signs and symptoms of TE (e.g., calf swelling and/or pain, shortness of breath), makes a substantial difference. However, there was no difference in effect estimates between studies that used a screening protocol and those that confirmed clinical signs and symptoms of TE.

While our data retrieval and extraction strategy complied with current methodological standards, multiple comparisons should be addressed by network meta-analysis or more advanced techniques. Again, the heterogeneity of the regimens prohibited this, and we feel that the presented bivariate estimates are sufficient for clinical and healthcare decision-making.

The present findings may only partially support the health–economic modelling of the utility and value of gCS in trauma and orthopaedic surgery. There is robust evidence that LMWH use in orthopaedic surgery is cost-effective [43,44]. The direct costs of LMWH and other substances, such as Fondaparinux, are far higher than those of gCS, but their effectiveness also exceeds that of gCS in preventing TE events. If gCS have no added value in pharmacological prophylaxis (as suggested by the present review), they must be abandoned, regardless of their relatively low direct costs. Including washing, it was estimated that they incur annual fees of about EUR 150,000 in our hospital group, which comprises nine tertiary-care trauma centres. A formal health–economic analysis, incorporating patients’ and their relatives’ preferences and health-related quality-of-life assessments, is needed to determine the utility of gCS. The reporting of gCS-related AE like skin issues, slipping, tolerability, etc., was insufficient across the included studies and represents a critical evidence gap.

## 5. Conclusions

Current evidence suggests that gCS do not provide a consistent additional benefit when used alongside pharmacological prophylaxis, although heterogeneity limits the strength of the conclusions.

## Figures and Tables

**Figure 1 jcm-14-08578-f001:**
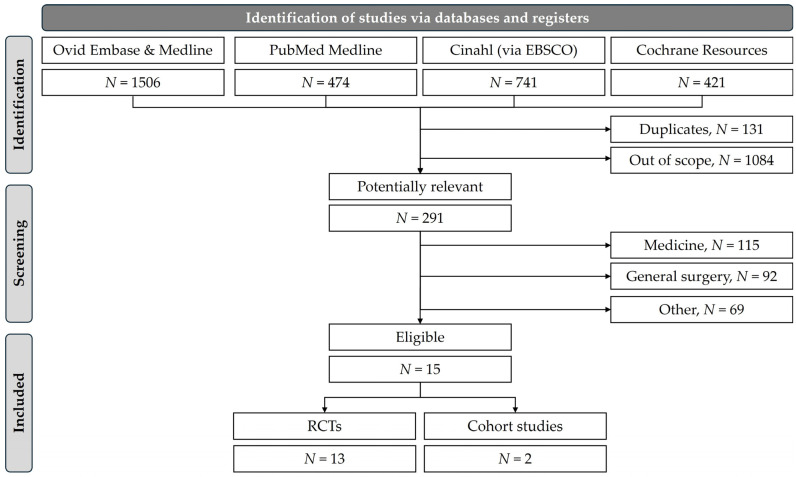
Flow diagram of the study selection process using PRISMA 2020.

**Figure 2 jcm-14-08578-f002:**
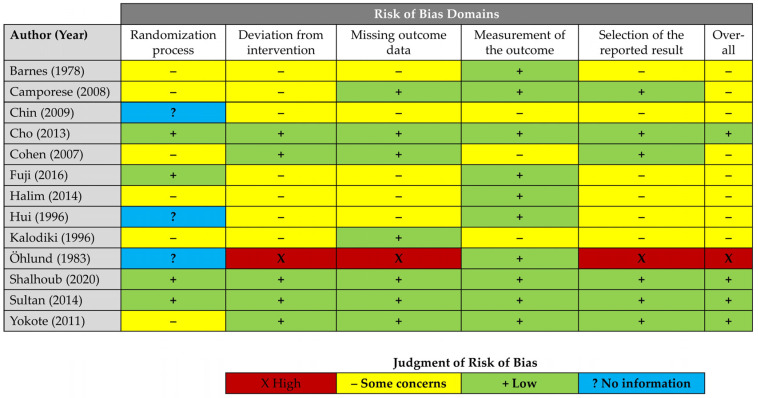
Risk of bias amongst RCTs as assessed by the Cochrane RoB-2 instrument [24,25,26,27,28,29,30,31,32,33,34,35,36].

**Figure 3 jcm-14-08578-f003:**
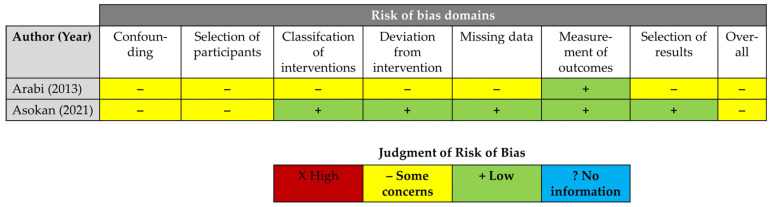
Risk of bias in observational studies according to the Cochrane ROBINS-I tool [37,38].

**Figure 4 jcm-14-08578-f004:**
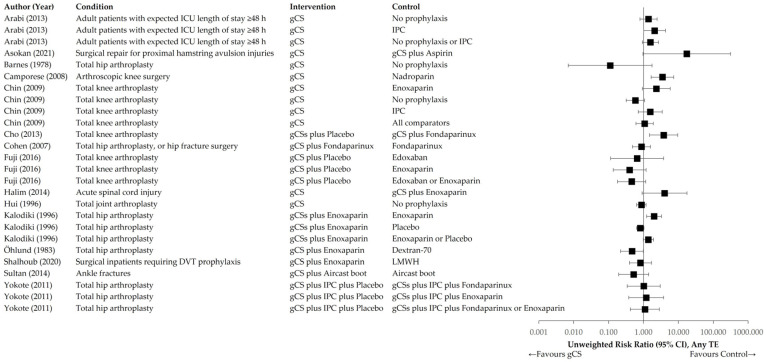
Unweighted RR of individual trials comparing gCS with any control intervention. Primary endpoint: any TE [24,25,26,27,28,29,30,31,32,33,34,35,36,37,38].

**Figure 5 jcm-14-08578-f005:**
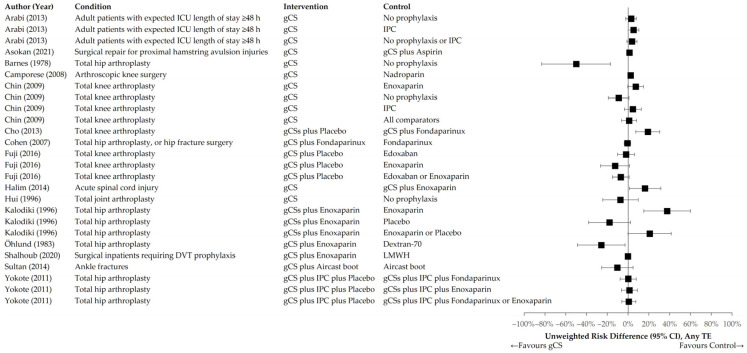
Unweighted RD of individual trials comparing gCS with any control intervention. Primary endpoint: any TE [24,25,26,27,28,29,30,31,32,33,34,35,36,37,38].

**Figure 6 jcm-14-08578-f006:**
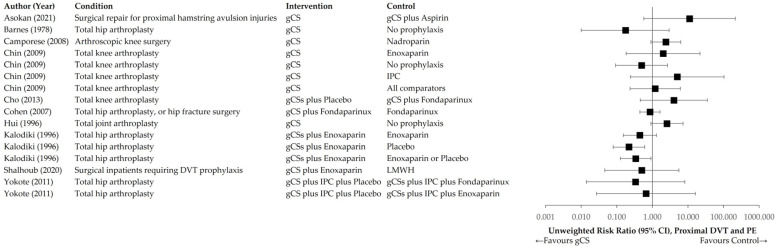
Unweighted RR of individual trials comparing gCS with any control intervention. Primary endpoint: proximal DVT or PE [24,25,26,27,30,31,33,35,36,38].

**Figure 7 jcm-14-08578-f007:**
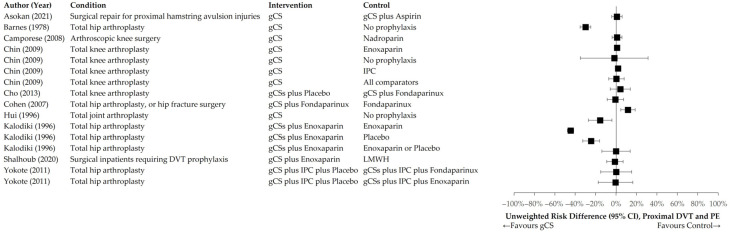
Unweighted RD of individual trials comparing gCS with any control intervention. Primary endpoint: proximal DVT or PE [24,25,26,27,30,31,33,35,36,38].

**Figure 8 jcm-14-08578-f008:**
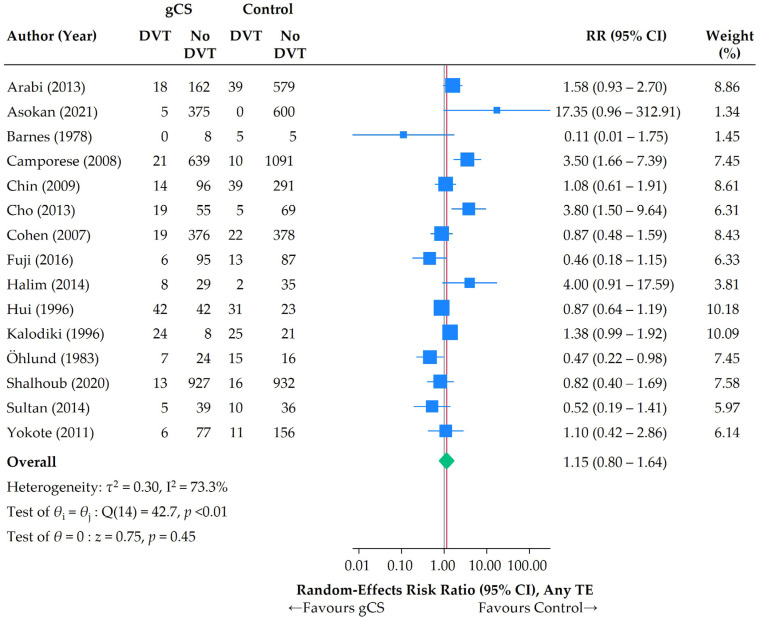
Random-effects meta-analysis of the effectiveness of gCS in different scenarios. Primary endpoint: any TE. Blue squares (sized according to the weight of studies in the meta-analysis), 95% confidence intervals, and green summary estimates are the default in STATA 18.0’s meta package [24,25,26,27,28,29,30,31,32,33,34,35,36,37,38].

**Figure 9 jcm-14-08578-f009:**
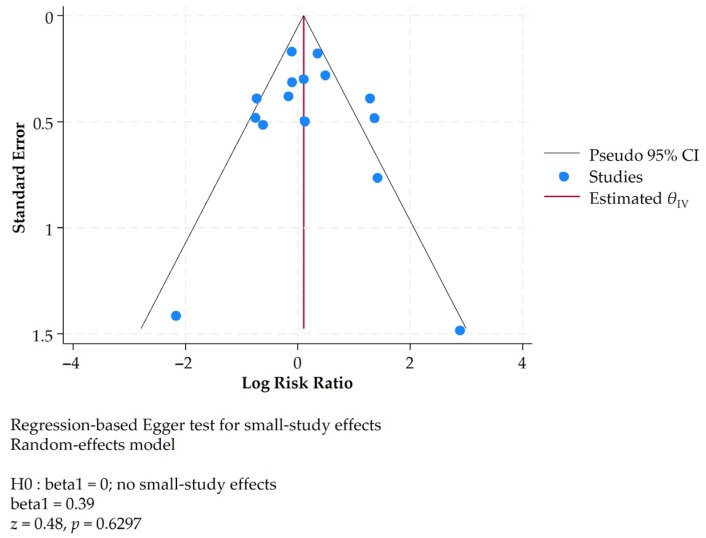
Funnel plot showing symmetry and, according to Egger’s test, no evidence of publication bias.

**Figure 10 jcm-14-08578-f010:**
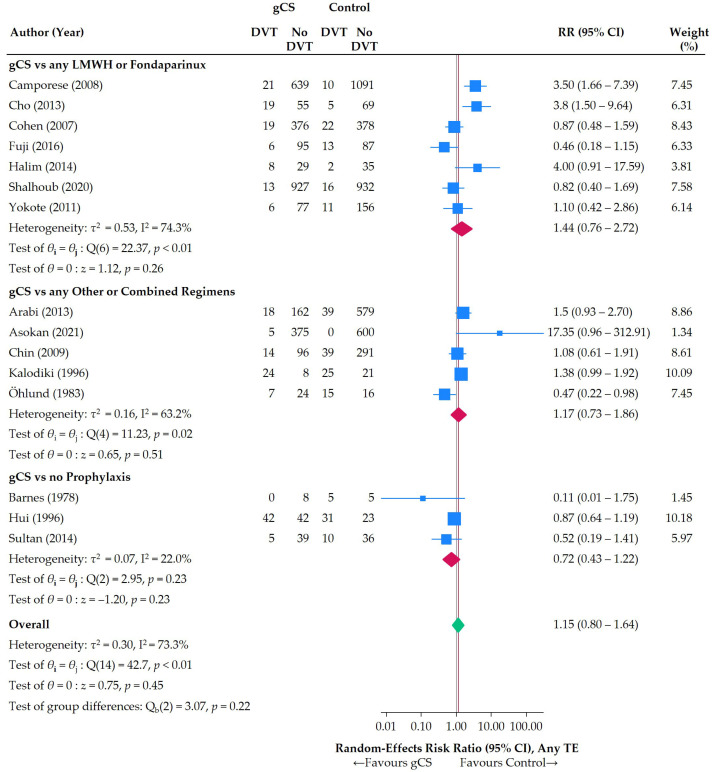
Individual and pooled RR, as stratified by treatment clusters. Blue squares (sized according to the weight of studies in the meta-analysis), 95% confidence intervals, and green summary estimates are the default in STATA 18.0’s meta package [24,25,26,27,28,29,30,31,32,33,34,35,36,37,38].

**Table 1 jcm-14-08578-t001:** Certainty of evidence according to GRADE. (S)AE = (serious) adverse events. CI = confidence interval. gCS = graduated compression stockings. HRQoL = health-related quality of life. RR = risk ratio. PE = pulmonary embolism. TE = thromboembolism. THA = total hip arthroplasty. TKA = total knee arthroplasty.

Should Graduated Compression Stockings Be Described to Lower the Incidence of Thromboembolic Events in Orthopaedic and/or Trauma Surgery?									
Outcome	Risk with gCS and/or Pharmacological Prophylaxis	Risk Without gCS and/or Pharmacological Prophylaxis	Relative Effect (RR, 95% CI)	No. of Participants Included	Certainty of the Evidence (GRADE)				Comments
Any event of TE	66 per 1000 (0.66%, 95% CI 0.57 to 0.75%)	53 per 1000 (0.53%, 95% CI 0.47 to 0.60%)	1.15 (95% CI, 0.80 to 1.64)	N = 7721	⊕	⊕	⊕	◯	Heterogeneity of interventions and diagnostic procedures ^1,2,3^, risk of partial verification bias ^4^
					Moderate				
Proximal DVT or PE	20 per 1000 (0.20%, 95% CI 0.15 to 0.26%)	16 per 1000 (0.16%, 95% CI 0.12 to 0.20%)	1.17 (95% CI, 0.62 to 2.21)	N = 6496	⊕	⊕	⊕	◯	Heterogeneity of interventions and diagnostic procedures ^1,2,3^, risk of partial verification bias ^4^
					Moderate				
Mortality									Inconsistent or insufficient data
AE/SAE									Inconsistent or insufficient data
HRQoL									Inconsistent or insufficient data
Function									Inconsistent or insufficient data

^1^ Direct comparison of gCS vs. no prophylaxis: Hui (1996) [30], TKA/THA, *N* = 140: no benefit; Barnes (1978) [24], THA, *N* = 18: favouring gCS; ^2^ Nine studies on total knee or hip arthroplasty, with or without hip replacement for femoral fractures; ^3^ Nine trials investigating LMWH or Fondaparinux with or without placebo control; ^4^ 13 trials claiming screening for DVT, but with different methods (e.g., different sonographic methods, chest CT etc.).

**Table 2 jcm-14-08578-t002:** Key characteristics of studies enrolled in this systematic review and meta-analysis.

Study	Country	Design	Condition	Diagnosis of TE	Proportion of Orthopaedic or Trauma Patients	Description of Modalities		Recruitment Period	*N*		Mean Age, Years		Males	
						Intervention	Control		Intervention	Control	Intervention	Control	Intervention	Control
Arabi (2013) [37]	Saudi Arabia	Prospective cohort with Propensity-score matching	Adult patients admitted to the ICU at King Abdulaziz Medical City in Riyadh, Saudi Arabia, with an expected length of stay of ≥48 h (trauma *N* = 141; pelvic or femur fractures *N* = 41)	At team’s discretion, Doppler ultrasonography, chest CT and/or ventilation perfusion scan	0.35	gCS	Intermittent pneumatic compression (*N* = 229) or no prophylaxis (*N* = 389)	07/2006–01/2008	180	618	49	52	130 (72%)	405 (63%)
Asokan (2021) [38]	UK	Prospective cohort	Complete or partial proximal hamstring avulsion injuries treated by tendon debridement and osseous re-attachment using 5.5 mm HEALIX Suture Anchors (DePuy Synthes)	Radiological investigation for VTE due to clinical suspicion	1.00	gCS	gCS plus aspirin: 150 mg once daily for 6 weeks	01/2000–12/2020	380	600	29	27	287 (76%)	479 (80%)
Barnes (1978) [24]	USA	RCT	Total hip arthroplasty (Charnley–Müller) via a modified Harris anterolateral approach	Screening Doppler ultrasonography	1.00	gCS	No prophylaxis	NS	8	10	64	68	2 (25%)	5 (50%)
Camporese (2008) [25]	Italy	RCT	Arthroscopic knee surgery (ACL reconstruction *N* = 598; any meniscectomy *N* = 704)	Screening Duplex ultrasonography	1.00	gCS	Nadroparin (3800 anti-Xa IU) for either 7 (*N* = 657) or 14 days (*N* = 444)	03/2002–01/2006	660	1101	42	42	398 (60%)	684 (62%)
Chin (2009) [26]	Singapore	RCT	Total knee arthroplasty	Screening Duplex ultrasonography	1.00	gCS	No prophylaxis (*N* = 110), intermittent pneumatic compression (*N* = 110), or Enoxaparin 40 mg once daily (*N* = 110)	01/2003–05/2004	110	330	67	66	14 (13%)	29 (9%)
Cho (2013) [27]	Republic of Korea	RCT	Total knee arthroplasty	Screening Duplex ultrasonography	1.00	Placebo (0.25 mL of isotonic saline) for five days, plus gCS	Fondaparinux (2.5 mg daily) for five days, plus gCS	11/2008–10/2010	74	74	69	69	7 (9%)	5 (7%)
Cohen (2007) [36]	UK, Brazil, Hong Kong, and Spain	RCT	Primary (*N* = 714) or revision (*N* = 42) total hip arthroplasty, or surgery for cervical (*N* = 26) or trochanteric (*N* = 13) fractures of the proximal femur	Unclear, ultrasonography, or venography (probably conditional on signs and symptoms)	1.00	Fondaparinux (2.5 mg daily) for five to nine days plus gCS for 35 to 49 days	Fondaparinux (2.5 mg daily) for five to nine days	01/2002–11/2004	391	404	65	65	163 (42%)	180 (45%)
Fuji (2016) [28]	Japan	RCT	Total knee arthroplasty	Screening venography	1.00	Edoxaban 30 mg orally once daily (*N* = 53) or Enoxaparin 20 mg subcutaneously twice daily (*N* = 48) plus gCS	Edoxaban 30 mg orally once daily (*N* = 52) or Enoxaparin 20 mg subcutaneously twice daily (*N* = 48)	10/2008–01/2010	101	100	73	71	17 (17%)	20 (20%)
Halim (2014) [29]	India	RCT	Acute spinal cord injury	Screening Duplex ultrasonography	1.00	gCS	Enoxaparin (40 mg) subcutaneously once a day, starting on the day of admission and continued for 8 weeks, plus gCS	12/2006–12/2010	37	37	NS	NS	35 (95%)	25 (68%)
Hui (1996) [30]	UK	RCT	Total hip arthroplasty (cemented Charnley, via a lateral approach), or total knee arthroplasty (PFC, Johnson & Johnson, 23% uncemented)	Screening venography	1.00	gCS (above knee *N* = 44, below knee *N* = 40)	No prophylaxis	NS	84	54	70	67	56 (67%)	33 (61%)
Kalodiki (1996) [31]	UK	RCT	Total hip arthroplasty	Screening venography	1.00	Enoxaparin (4000 anti Xa IU once daily) plus gCS	Enoxaparin (4000 anti Xa IU once daily *N* = 32) only or Placebo (*N* = 14)	NS	32	46	69	70	19 (59%)	19 (43%)
Öhlund (1983) [32]	Sweden	RCT	Total hip arthroplasty	Fibrinogen uptake test	1.00	Dextran-70 plus gCS	Dextran-70	11/1978–05/1979	31	31	NS	NS	30/62 males (48%)	
Shalhoub (2020) [33]	UK	RCT	Surgical inpatients requiring pharmaco-prophylaxis (orthopaedic surgery *N* = 28, neurosurgery *N* = 62, plastic surgery *N* = 39)	Screening Duplex ultrasonography	0.07	gCS plus LMWH	LMWH	05/2016–01/2019	940	948	58	59	346 (37%)	347 (37%)
Sultan (2014) [34]	UK	RCT	Ankle fractures (Weber A *N* = 36, B *N* = 42, C *N* = 12, ORIF *N* = 30)	Screening Duplex ultrasonography	1.00	Aircast boot (DJO Global, Vista, California) plus Ankle injury stockings (AIS)	Aircast boot (DJO Global, Vista, California) plus Tubigrip (Mölnlycke Health Care, Gothenburg, Sweden)	18 months	44	46	46	47	19 (43%)	17 (37%)
Yokote (2011) [35]	Japan	RCT	Total hip arthroplasty via an anterolateral modified Watson–Jones approach (cementless S-ROM-A, DePuy, CentPillar, Stryker, Taperloc, Biomet, or Versys, Zimmer)	Screening Duplex ultrasonography	1.00	gCS plus intermittent pneumatic compression plus subcutaneous placebo (0.5 mL of isotonic saline) for ten days	gCS plus intermittent pneumatic compression plus subcutaneous fondaparinux (2.5 mg once daily) or enoxaparin (20 mg twice daily) for ten days	05/2008–03/2009	83	167	63	64	16 (19%)	30 (18%)

**Table 3 jcm-14-08578-t003:** Summary of subset and sensitivity analyses. *k* = number of studies. *N* = number of participants. TE = Thromboembolism. DVT = deep vein thrombosis. PE = pulmonary embolism.

	Primary Endpoint: Any TE				Secondary Endpoint: Proximal DVT or PE			
Subgroup	*k*	*N*	RR (95% CI)	RD (95% CI)	*k*	*N*	RR (95% CI)	RD (95% CI)
Design								
RCT	13	5943	1.07 (0.73–1.57)	−0.5% (−6.6–5.6%)	9	5516	1.07 (0.56–2.02)	0.0% (−0.5–0.6%)
Cohort study	2	1778	3.38 (0.38–29.94)	1.4% (0.3–2.6%)	1	980	11.04 (0.57–213.17)	0.8% (−0.1–1.7%)
TE diagnosis								
Screening	12	5148	1.09 (0.71–1.67)	−0.8% (−8.3–6.7%)	8	4721	1.12 (0.51–2.45)	0.0% (−0.5–0.7%)
Discretionary	3	2573	1.36 (0.70–2.62)	1.1% (0.1–2.2%)	2	1775	2.01 (0.19–21.44)	0.1% (−0.1–0.1%)
Indication								
Arthroplasty	9	2130	0.96 (0.66–1.39)	−2.7% (−13.2–7.7%)	7	1867	0.95 (0.45–1.98)	−0.4% (−1.4–0.6%)
Other	6	5591	1.70 (0.81–3.58)	1.3% (−0.2–2.7%)	3	4629	2.22 (0.95–5.22)	0.4% (−0.3–1.1%)
Publication year								
<2000	4	296	0.81 (0.45–1.46)	−14.2% (−42.4–14.0%)	3	234	0.67 (0.13–3.37)	−12.3% (−39.2–14.5%)
≥2000	11	7425	1.35 (0.86–2.13)	1.0% (−0.3–2.3%)	7	6262	1.40 (0.75–2.63)	0.2% (−0.3–0.8%)
Population								
Ortho/Trauma	13	5035	1.50 (0.74–1.77)	−0.3% (−6.4–5.8%)	8	6228	1.23 (0.63–2.42)	0.4% (−0.4–1.1%)
Mixed	2	2686	1.20 (0.63–2.27)	0.1% (-2.7–4.6%)	2	268	0.50 (0.05–5.55)	−0.1% (−0.4–0.3%)

## Data Availability

Data abstracted from individual articles will be made available to other researchers upon request from the authors. The authors have reviewed and edited the output and take full responsibility for the content of this publication.

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
