# Peer review of "Graduated Compression Stockings for Thromboprophylaxis in Orthopaedic and Trauma Surgery: A Rapid Review and Meta-Analysis"

_jcm, 2025, doi:10.3390/jcm14238578_

Round 1

Reviewer 1 Report

Comments and Suggestions for Authors

Well done meta analysis of available data. Presented the discussion well. Nuanced take on gCS and their role. Adds to the scientific literature and clarifies the value of a common practice. 

My only concern is with the start of the discussion. Especially the second paragraph. Few of the claims do not align with standard orthopaedic practice nor were they supported in the cited studies . It is not often thought that graduated compression stockings are a confounder for infection as alluded in Line 256. gCS are often omitted in cases of Below knee pathologies with large dressings or more typically in splints. The only Below knee study included in your analysis was the Sultan paper in citation 27. Even in that study majority of patients were treated conservatively and wouldn't have the alluded issues with compliance due to wound care. 

Would reframe the start of the discussion with more appropriate orthopaedic practice 

Author Response

Reviewer #1

Well done meta-analysis of available data. Presented the discussion well. Nuanced take on gCS and their role. Adds to the scientific literature and clarifies the value of a common practice. 

We thank Reviewer #1 for his /her kind and motivating words, greatly appreciated!

It is not often thought that graduated compression stockings are a confounder for infection as alluded in Line 256. 

We fully agree with the comments, and deleted passages on any infection risk with gCS entirely.

Would reframe the start of the discussion with more appropriate orthopaedic practice.

We rewrote the discussion section, hoping to address this reviewer's concerns.

Again, we thank you for your time and efforts spent on our behalf!

Reviewer 2 Report

Comments and Suggestions for Authors

This study evaluates the efficacy of graduated compression stockings (gCS) for thromboprophylaxis in orthopedic and trauma surgery via a systematic review and meta-analysis, concluding that gCS provide no significant additional benefit beyond pharmacological prophylaxis. While the methodology is generally rigorous, the study has limitations including high heterogeneity, insufficient reporting of key outcomes, and a need for more precise definition of its innovation. Specific comments are as follows:

  1. High heterogeneity of included studies with inadequate source analysis: The included studies span a wide time range (1978–2021), involve outdated interventions (e.g., Dextran-70), and show significant variations in surgical types (total joint arthroplasty/spine/pelvis) and patient risk stratification, leading to high heterogeneity (I² = 71%). Additional subgroup analyses are recommended.
  2. Lack of discussion from a health economic perspective: Costs related to gCS and pharmacological agents are critical for clinical resource allocation, yet no such data are reported in the included studies. It is recommended to explicitly address this limitation in the discussion.
  3. Need for more rigorous definition of innovation: There have been numerous prior studies on gCS for thromboprophylaxis in orthopedic surgery (e.g., doi:10.1016/j.jvsv.2018.05.020;doi: 10.1186/s43019-022-00166-y;doi: 10.1186/s43019-024-00250-5).
  4. Insufficient reporting of adverse events (AEs): The included studies inadequately report gCS-related skin AEs and drug-related bleeding events. It is recommended to emphasize this limitation in the discussion.
  5. Sensitivity analyses: Some early RCTs have extremely small sample sizes (e.g., Barnes 1978), which may compromise result stability. Additional sensitivity analyses are recommended.

Author Response

Reviewer #2

This study evaluates the efficacy of graduated compression stockings (gCS) for thromboprophylaxis in orthopedic and trauma surgery via a systematic review and meta-analysis, concluding that gCS provide no significant additional benefit beyond pharmacological prophylaxis. While the methodology is generally rigorous, the study has limitations including high heterogeneity, insufficient reporting of key outcomes, and a need for more precise definition of its innovation.

We thank Reviewer #2 for her/his motivating words and excellent suggestions.

High heterogeneity of included studies with inadequate source analysis: The included studies span a wide time range (1978–2021), involve outdated interventions (e.g., Dextran-70), and show significant variations in surgical types (total joint arthroplasty/spine/pelvis) and patient risk stratification, leading to high heterogeneity (I² = 71%). Additional subgroup analyses are recommended.

Greatly appreciated. We added multiple subgroup analyses, including studies published before and after 2000. We had done meta-regression, but felt that this approach did not is not informative to readers. We included a subset analysis of studies published before and after 2000 (without remarkable changes in effect estimates, i.e., RR and RD).

Lack of discussion from a health economic perspective: Costs related to gCS and pharmacological agents are critical for clinical resource allocation, yet no such data are reported in the included studies. It is recommended to explicitly address this limitation in the discussion.

Done. We are grateful for this comment. LMWH may be dominant in this setting, and gCS hardly contribute anything to the risk of TE, proximal DVT, or PE. We added a paragraph to the discussion section.

Need for more rigorous definition of innovation: There have been numerous prior studies on gCS for thromboprophylaxis in orthopedic surgery (e.g., doi:10.1016/j.jvsv.2018.05.020;doi: 10.1186/s43019-022-00166-y;doi: 10.1186/s43019-024-00250-5).

Done as well. We cited and briefly explained the mentioned studies, and stressed how the present work add to previous data.

Insufficient reporting of adverse events (AEs): The included studies inadequately report gCS-related skin AEs and drug-related bleeding events. It is recommended to emphasize this limitation in the discussion.

Hard but good point. We incorporated it in the new subsection 3.3. We hope our response is good enough.

Sensitivity analyses: Some early RCTs have extremely small sample sizes (e.g., Barnes 1978), which may compromise result stability. Additional sensitivity analyses are recommended.

Indeed, the problem with the almost 50 year old trial by Barnes is not (only) the small sample size, but interventions which are now outdated. A subset analysis respected this (see new Table 3).

We hope we could address all concerns of Reviewer #2 to his/her complete satisfaction.

Reviewer 3 Report

Comments and Suggestions for Authors

The manuscript presents a systematic review and meta-analysis assessing the effectiveness of graduated compression stockings (gCS) for venous thromboembolism (VTE) prophylaxis in orthopedic and trauma surgery. The authors followed PRISMA guidelines, conducted a structured literature search, and performed quantitative pooling of outcomes related to deep vein thrombosis (DVT) and pulmonary embolism (PE).

The topic is clinically relevant and timely, given the persistent uncertainty regarding the additive value of mechanical prophylaxis in the era of routine pharmacologic thromboprophylaxis and enhanced-recovery protocols. The review is well-organized and generally clear, but several methodological and interpretive aspects should be expanded to strengthen both its scientific rigor and its clinical applicability.

  1. A clearly defined evidence gap would enhance the paper’s originality and help readers, understand the specific need for this new meta-analysis. The introduction should therefore provide a more explicit justification of the study’s necessity in relation to prior reviews (e.g., Cochrane 2017, NICE 2021, ACCP Guidelines). Currently, the rationale emphasizes the overall importance of thromboprophylaxis but does not specify what previous analyses may have overlooked, such as the exclusion of trauma populations, the lack of stratification by surgical type, or the omission of data reflecting more recent pharmacologic protocols.
  2. A number of the included studies seem to draw from quite different surgical populations (a mix of orthopedic, trauma, and even some elective procedures). It would help if the authors explained how data from non-orthopedic cases were handled in the analysis. Were these studies excluded, analyzed separately, or proportionally weighted? Combining such diverse groups can easily inflate statistical heterogeneity and blur meaningful subgroup patterns. Running a sensitivity analysis restricted to clearly orthopedic or trauma cohorts would make the main conclusions more robust and give readers confidence that the results are not being skewed by population differences.
  3. The search strategy appears comprehensive, but a few practical details are missing that would help readers judge how reproducible the process really is. It would be useful to specify how duplicate records were managed, how non–peer-reviewed sources were treated, and whether backward or forward citation tracking (for example, through Scopus or Web of Science) was performed. Explaining these steps would make the methods more transparent and bring the review fully in line with PRISMA 2020 standards. It would also be helpful to note the exact date of the last database search and mention any differences from the registered PROSPERO protocol, as these clarifications add credibility and show careful methodological control—points editors and readers both appreciate in evidence syntheses.
  4. The authors have used an appropriate risk-of-bias tool, but the review would benefit from adding a clear GRADE assessment that summarizes the overall certainty of evidence for the main outcomes—DVT and PE. Including even a brief “Summary of Findings” table would make the results easier to interpret and highlight how much confidence readers can place in the conclusions. To my knowledge, GRADE tables have become standard in meta-analyses aimed at informing clinical practice because they link effect size with the quality of evidence, allowing readers to see both the statistical and the clinical significance at a glance. This addition would also align the paper more closely with PRISMA recommendations for transparent reporting.
  5. The reported heterogeneity (I² = 71%) is quite high but not discussed in enough depth. The discussion would be stronger if it explored what might be driving this variation, such as differences in pharmacologic prophylaxis, the era in which studies were conducted, surgical type, or regional practice patterns. If feasible, a meta-regression or a stratified subgroup analysis (for example, hip versus knee versus trauma surgery) could help clarify these effects. Even a brief acknowledgment of these factors as limitations would add valuable context.
    Addressing heterogeneity is important to interpreting pooled results; without this, readers may struggle to judge whether the reported “no significant effect” truly reflects an absence of efficacy or simply the influence of mixed protocols. A more nuanced exploration would give the conclusions greater credibility and reassure readers about the robustness of the findings.
  6. The conclusion that “current best evidence does not support the use of gCS…” comes across as a little too definitive considering the limited number of high-quality trials and the substantial variability between studies. It might read more accurately if the authors acknowledged the uncertainty that still exists in this area. A balanced alternative could be: “Current evidence suggests that graduated compression stockings provide no consistent additional benefit when used alongside pharmacologic thromboprophylaxis, although study heterogeneity limits the strength of any conclusions.”

This way it would prevent the impression of overstatement and bring the discussion more in line with the journal’s preference for an even-handed, evidence-based interpretation. It could also be worthwhile to include a short paragraph addressing patient-centered factors like comfort, adherence, cost, and possible skin irritation, which often shape real-world decision-making. Finally, highlighting potential subgroups that might still benefit, such as high-risk trauma patients or those who cannot receive anticoagulation, would add nuance and make the findings more clinically relevant.

Author Response

The manuscript presents a systematic review and meta-analysis assessing the effectiveness of graduated compression stockings (gCS) for venous thromboembolism (VTE) prophylaxis in orthopedic and trauma surgery. The authors followed PRISMA guidelines, conducted a structured literature search, and performed quantitative pooling of outcomes related to deep vein thrombosis (DVT) and pulmonary embolism (PE). The topic is clinically relevant and timely, given the persistent uncertainty regarding the additive value of mechanical prophylaxis in the era of routine pharmacologic thromboprophylaxis and enhanced-recovery protocols. The review is well-organized and generally clear, but several methodological and interpretive aspects should be expanded to strengthen both its scientific rigor and its clinical applicability.

We thank Reviewer #3 for her/his nice words, and are more than happy to address the concerns.

A clearly defined evidence gap would enhance the paper’s originality and help readers, understand the specific need for this new meta-analysis. The introduction should therefore provide a more explicit justification of the study’s necessity in relation to prior reviews (e.g., Cochrane 2017, NICE 2021, ACCP Guidelines). Currently, the rationale emphasizes the overall importance of thromboprophylaxis but does not specify what previous analyses may have overlooked, such as the exclusion of trauma populations, the lack of stratification by surgical type, or the omission of data reflecting more recent pharmacologic protocols.

We referenced previous systematic reviews mentioned by other experts, and how our work differs from previous ones. We like to stress that we focus on orthopaedics and trauma, and that the role of gCS has not yet been studied in such a way.

A number of the included studies seem to draw from quite different surgical populations (a mix of orthopedic, trauma, and even some elective procedures). It would help if the authors explained how data from non-orthopedic cases were handled in the analysis. Were these studies excluded, analyzed separately, or proportionally weighted? Combining such diverse groups can easily inflate statistical heterogeneity and blur meaningful subgroup patterns. Running a sensitivity analysis restricted to clearly orthopedic or trauma cohorts would make the main conclusions more robust and give readers confidence that the results are not being skewed by population differences.

Excellent point! We did multiple extra subset and sensitivity analyses (see Table 3), and tried to be as "orthopaedic- and trauma-specific" as we could. For example, the GAPS trial (probably the current best and largest evaluating the added effect of gCS to TE prophylaxis by LWMH in surgical inpatients) was excluded in a subset analysis because no precise information could be derived even from the NHR report.

The search strategy appears comprehensive, but a few practical details are missing that would help readers judge how reproducible the process really is. It would be useful to specify how duplicate records were managed, how non–peer-reviewed sources were treated, and whether backward or forward citation tracking (for example, through Scopus or Web of Science) was performed. Explaining these steps would make the methods more transparent and bring the review fully in line with PRISMA 2020 standards. It would also be helpful to note the exact date of the last database search and mention any differences from the registered PROSPERO protocol, as these clarifications add credibility and show careful methodological control—points editors and readers both appreciate in evidence syntheses.

We thank Reviewer #3 for mentioning this point. As we compiled all citations in EndNote, we excluded all duplicates right away. We also stated that our latest search was conducted on March 01, 2025. We cannot comment on the issue of backward and forward tracking here, as it is beyond the scope of this contribution (https://libguides.fau.edu/c.php?g=325509&p=2182112, Hirt J, Nordhausen T, Appenzeller-Herzog C, Ewald H. Citation tracking for systematic literature searching: A scoping review. Res Syn Meth. 2023; 14(3): 563-579. doi:10.1002/jrsm.1635). The latest paper included in our analysis was published 2021.

The authors have used an appropriate risk-of-bias tool, but the review would benefit from adding a clear GRADE assessment that summarizes the overall certainty of evidence for the main outcomes—DVT and PE. Including even a brief “Summary of Findings” table would make the results easier to interpret and highlight how much confidence readers can place in the conclusions. To my knowledge, GRADE tables have become standard in meta-analyses aimed at informing clinical practice because they link effect size with the quality of evidence, allowing readers to see both the statistical and the clinical significance at a glance. This addition would also align the paper more closely with PRISMA recommendations for transparent reporting.

Excellent point. We added a GRADE summary table, according to https://www.gradepro.org/. 

The reported heterogeneity (I² = 71%) is quite high but not discussed in enough depth. The discussion would be stronger if it explored what might be driving this variation, such as differences in pharmacologic prophylaxis, the era in which studies were conducted, surgical type, or regional practice patterns. If feasible, a meta-regression or a stratified subgroup analysis (for example, hip versus knee versus trauma surgery) could help clarify these effects. Even a brief acknowledgment of these factors as limitations would add valuable context.
Addressing heterogeneity is important to interpreting pooled results; without this, readers may struggle to judge whether the reported “no significant effect” truly reflects an absence of efficacy or simply the influence of mixed protocols. A more nuanced exploration would give the conclusions greater credibility and reassure readers about the robustness of the findings.

We conducted multiple additional subgroup analyses to explain the substantial heterogeneity.

The conclusion that “current best evidence does not support the use of gCS…” comes across as a little too definitive considering the limited number of high-quality trials and the substantial variability between studies. It might read more accurately if the authors acknowledged the uncertainty that still exists in this area. A balanced alternative could be: “Current evidence suggests that graduated compression stockings provide no consistent additional benefit when used alongside pharmacologic thromboprophylaxis, although study heterogeneity limits the strength of any conclusions.”

We changed the wording to "Current best evidence does not support the routine use of graduated compression stockings to prevent thromboembolism in orthopaedic and trauma surgery if patients are under pharmacological prophylaxis by low-molecular-weight heparins, fondaparinux, or other agents proven to inhibit the intrinsic coagulation cascade."

We hope we were able to address all concerns of Reviewer #3 appropriately, and that he/she supports us in publishing our manuscript.

Round 2

Reviewer 3 Report

Comments and Suggestions for Authors

The authors have thoroughly and satisfactorily addressed all concerns raised in the first review. The manuscript is now clearer, more rigorous, and better aligned with methodological standards for systematic reviews and meta-analyses. The additions of subgroup analyses, GRADE certainty assessment, and expanded methodological transparency significantly strengthen the paper.

Author Response

Dear Editors, Dear JCM Team and Reviewers,

we are more than pleased and delighted our work was considered eligible for publication! 

We are grateful for your advice! As suggested, we added the following parts to the discussion section:

"In addition, differences in pharmacological protocols, methods for detecting thromboembolism, study eras, and various surgical conditions contributed to the dispersion of effect sizes."

"A formal health-economic analysis, incorporating patients’ and their relatives’ preferences and health-related quality-of-life assessments, is needed to determine the utility of gCS. Reporting of gCS-related AE like skin issues, slipping, tolerability etc. was insufficient across included studies and represents a critical evidence gap."

The Conclusion now reads:

"Current evidence suggests that gCS do not provide a consistent additional benefit when used alongside pharmacological prophylaxis, although heterogeneity limits the strength of conclusions."

Again, we thank you for your valuable comments and suggestions, and look forward to the next steps of publication.

Kind regards

Dirk Stengel, MD PhD MSc